# The Impact of Caloric Restriction on the Epigenetic Signatures of Aging

**DOI:** 10.3390/ijms20082022

**Published:** 2019-04-24

**Authors:** Noémie Gensous, Claudio Franceschi, Aurelia Santoro, Maddalena Milazzo, Paolo Garagnani, Maria Giulia Bacalini

**Affiliations:** 1Department of Experimental, Diagnostic and Specialty Medicine (DIMES), University of Bologna, 40126 Bologna, Italy; noemie_gensous@hotmail.com (N.G.); aurelia.santoro@unibo.it (A.S.); maddimilazzo@hotmail.it (M.M.); 2IRCCS Istituto delle Scienze Neurologiche di Bologna, 40139 Bologna, Italy; claudio.franceschi@unibo.it (C.F.); mariagiuli.bacalini2@unibo.it (M.G.B.); 3Lobachevsky State University of Nizhny Novgorod, Nizhny Novgorod 603022, Russia; 4Center for Applied Biomedical Research (CRBA), St. Orsola-Malpighi University Hospital, 40138 Bologna, Italy; 5Clinical Chemistry, Department of Laboratory Medicine, Karolinska Institutet at Huddinge University Hospital, S-141 86 Stockholm, Sweden; 6Institute of Molecular Genetics (IGM)-CNR, Unit of Bologna, 40136 Bologna, Italy; 7Laboratory of Musculoskeletal Cell Biology, Rizzoli Orthopedic Institute, 40136 Bologna, Italy

**Keywords:** aging, nutrition, caloric restriction, epigenetic clocks

## Abstract

Aging is characterized by an extensive remodeling of epigenetic patterns, which has been implicated in the physiopathology of age-related diseases. Nutrition plays a significant role in modulating the epigenome, and a growing amount of data indicate that dietary changes can modify the epigenetic marks associated with aging. In this review, we will assess the current advances in the relationship between caloric restriction, a proven anti-aging intervention, and epigenetic signatures of aging. We will specifically discuss the impact of caloric restriction on epigenetic regulation and how some of the favorable effects of caloric restriction on lifespan and healthspan could be mediated by epigenetic modifications.

## 1. Introduction

Aging can be defined as a time-dependent and progressive decline in functional status which ultimately results in death. This complex and multifaceted phenomenon, which is a major risk factor for a wide range of non-communicable and chronic diseases, is the result of a complex interplay between genetic, environmental, and stochastic variables. While some researchers have suggested that aging is genetically programmed, others sustain that it is the result of the progressive accumulation of damage coupled with a decline in maintenance [1,2,3]. While this topic is still controversial, the scientific community agrees that aging is characterized by specific hallmarks, which rely on well-defined biological pathways [4,5]. The identification of the mechanistic processes underlying aging has one important consequence: it provides new potentially modifiable targets and justifies anti-aging interventions.

Caloric restriction (CR), i.e., the reduction of caloric intake (by 10% to 40%) without causing malnutrition, has proven to be by far the most effective intervention that can extend the maximum lifespan in a wide range of organisms including yeast, nematodes, flies, and rodents [6]. Interestingly, observations also demonstrated an effect on healthspan (i.e., time spent being healthy), coincident with a significant decrease in age-related diseases such as cardiovascular events, diabetes, neurodegenerative diseases, and cancers [7,8,9,10]. The beneficial effects of CR occur through an extreme wide range of molecular mechanisms, largely overlapping with aging hallmarks, among which epigenetic factors have recently gained interest.

The three main categories of epigenetic marks, DNA methylation (DNAm), post-translational histone modifications (PTHMs) and small non-coding RNAs, are considered as central regulators of the aging process [4,5,11,12]. Epigenetics patterns are greatly remodeled during aging, and this phenomenon has been implicated in the development of multiple age-related diseases [13,14]. One important characteristic of epigenetic factors is their plasticity: epigenetic changes are reversible and they can be modulated by a wide range of environmental factors, including lifestyle habits and dietary interventions [15]. 

In this article, we will specifically review the current knowledge on the impact of CR on the epigenetic signatures of aging, discussing also the effect on the recently developed DNA methylation clocks. 

## 2. DNA Methylation

DNA methylation consists in the addition of a methyl group to a cytosine residue, preferentially in a CpG dinucleotide. During aging, there is a marked remodeling of genomic DNA methylation patterns, which has been extensively investigated and implicated in multiple common age-related diseases [13,14,16,17]. Different types of DNA methylation changes coexist during aging. Firstly, the global genomic DNA methylation level decreases with age, mainly reflecting a decrease in DNA methylation of repetitive genomic regions and interspersed elements [18,19]. Secondly, differential methylation of specific genomic loci occurs, especially with an age-related hypermethylation in gene promoters [20,21,22,23]. Finally, there is an increase in the inter-individual divergence between patterns of DNA methylation (named DNA methylation drift) [24,25,26,27], and an increase in the rate of epimutations [28]. 

The two first studies reporting the impact of CR on DNA methylation were published in 1993: CR was shown to induce an increase in the methylation of the proto-oncogenes *c-myc* [29] and *Ras* [30] in mice livers and pancreatic acinar rat cells, respectively, repressing the trend towards age-associated hypomethylation [29,30]. Twenty years later, Ions et al. analyzed the impact of CR on a genome-wide scale and correlated epigenetic data with transcriptomic ones [31]. Indeed, it has been described that CR impacts gene expression across numerous organisms, preventing changes observed in the aging transcriptomes [32,33,34,35,36,37]. Analyzing publicly available datasets, Ions et al. reported a significant overlap between the genes that showed altered expression in response to CR and those whose methylation varies during aging [31]. 

Recently, nine independent studies comparing the effect of age on DNA methylation patterns in animals fed ad libitum (AL) or with a CR diet have been published [25,38,39,40,41,42,43,44,45]. While CR seemed to have no impact on DNA methylation levels in adult *Drosophila* [44], eight other studies collectively reported that CR is protective against age-related DNA methylation changes in mammals in different tissues types (kidney [38], blood [25,40], liver [39,42], hippocampus [41], and cerebellum [45]). Genomic regions which tend to become differentially methylated with age (aDMRs) experience less changes in animals under a CR diet: for example, analyzing the liver from female mice on a CR diet from 4 to 22 months of age, Cole et al. observed that CR increased methylation in hypomethylated aDMRs, while it decreased it in the hypermethylated aDMRs. Interestingly, Sziraki et al. observed a two-stage response to CR [40]. They analyzed the blood methylomes of four groups of mice, aged from 10 to 27 months, which all started CR at the age of 4 months. They found that the DNA methylome was initially shifted by CR in the same direction as aging but, secondarily, the cumulative changes associated with CR shifted it toward a younger state compared to control animals [40]. The remodeling of DNA methylation patterns associated with CR can target genomic regions associated with the development of age-related diseases. For example, in the kidney of old rats, CR was able to attenuate age-dependent methylation alterations in the promoters of genes that are associated with inflammation, cancer, or diabetes [38], while in mouse liver, CR had a specific impact on genes involved in lipid metabolism-related pathways, resulting in the regulation of the lipid profile (with an attenuation of the age-associated increase in liver triglyceride content) [39].

While the previously mentioned studies evaluated the impact of CR on DNA methylation directional changes (hypo- or hypermethylation), Maegawa et al. investigated its influence on DNA methylation drift. This progressive divergence of the epigenomes between different subjects over time has been correlated with lifespan in three different mammalian species (mouse, rhesus monkey, and human) [25,46]. Maegawa et al. demonstrated that CR was able to protect against this DNA methylation drift, both in mice and rhesus macaques. The observed effects were possibly dose-dependent, regarding both CR severity and duration: indeed, in monkeys exposed to 30% CR since middle age, the attenuation of age-related methylation drift as compared to AL fed controls was less pronounced than the one seen in mice exposed to 40% CR since early adulthood [25]. 

Interestingly, an important characteristic of CR is its capacity to induce a cellular memory which can persist even when it is discontinued [47,48,49]. DNA methylation could play a role in these long-lasting effects. It was demonstrated that even a short-term CR (one or two months) can reverse changes in aDMRs in rodents [38,39]. In four-month-old male mice, a one-month CR was able to induce significant changes in the expression of several genes, and 20 to 50% of these changes persisted two months after CR was discontinued [50]. Interestingly, concomitant significant changes in DNA methylation of the promoter regions of those genes were observed. For example, CR induced a significant decrease in the DNA methylation of *Nts1* gene promoter, correlating with the increase in its mRNA expression. The hypomethylation persisted even when CR was discontinued [50]. Long-lasting effects of severe dietary restriction on epigenetics have also been observed in humans exposed prenatally to famine during the Dutch Hunger Winter in 1944–1945. Remarkably, it was observed that differential methylation persisted even after six decades [51]. 

Epigenetic data on the effects of pure CR in humans are limited, as it is an intervention likely to be difficult to implement in the long-run in humans. To our knowledge, only two studies have reported the results of epigenomic responses to a hypocaloric diet intervention in humans [52,53], but no study has specifically evaluated the impact of CR on DNA methylation signatures of aging in humans. A reduced energy intake intervention was shown to cause DNA methylation changes at specific loci in obese subjects (three chromosomal regions (chromosomes 1p36, 4q21, and 5q13) in Bouchard et al. study; *ATP10A* and *WT1* genes in Milagro et al. study) [52,53]. According to these studies, DNA methylation levels could also be used as predictors of weight loss, as they were significantly different between high and low responders to the nutritional intervention [52,53,54]. 

Age-related changes in DNA methylation patterns can be partially related to modifications in the expression and/or activity of enzymes which catalyze the addition of the methyl group on DNA, i.e., DNA methyltransferases (DNMTs). At least three independent DNMTs are usually involved in the establishment and maintenance of DNA methylation profiles: DNMT1 is responsible for maintenance of DNA methylation, while DNMT3a and DNMT3b act as de novo methyltransferases. During aging, a decrease in the activity and in the expression of DNMT1 has been observed [55,56]. Exposition to glucose restriction in vitro induced an increase in DNMT1 activity in normal fetal lung fibroblasts [57], thus counteracting age-related changes. On the other side, CR was able to antagonize the age-related increase in the levels of DNMT3a in mouse hippocampus, especially in the hippocampal dentate gyrus region [58]. 

## 3. miRNAs

Another mechanism of epigenetic regulation is represented by noncoding RNAs, among which microRNAs (miRNAs) are the best characterized. miRNAs are a broad class of small non-coding RNAs that exert key roles in the post-transcriptional regulation of gene expression by targeting specific mRNAs. Recent studies, performed in different species, have shown that numerous miRNAs are differentially expressed during aging [59,60,61,62]. 

It has been demonstrated that CR was able to impact the age-related differential expression of miRNAs, in different tissues (Table 1). For example, in the brain of mice, CR was able to counteract the age-dependent increase in miR-181a-1*, miR-30e, and miR-34a, leading to a gain in the expression of their common target gene, *Bcl-2*, involved in apoptosis [63]. The persistent elevated levels of *Bcl-2* in response to CR in the brains of older animals could decrease apoptosis and induce in this way a gain in neuronal survival, participating in the neuroprotective effects of CR [64]. Additionally, in rat brains, Wood et al. also identified differentially expressed miRNAs with age and CR. One of them, miR-98-3p, which is implicated in the modification of histone deacetylase (HDAC) and histone acetyltransferase (HAT) activities, was consistently overexpressed with CR [35]. In skeletal muscle, using RNA-sequencing to characterize the profiles of young and old rhesus monkeys, it was also demonstrated that CR is able to reverse the age-related alterations in miRNA expression towards a younger phenotype [65]. Finally, in mice serum, Dhahbi et al. observed an age-related increase in the levels of many circulating miRNAs, increase which was antagonized by long-term CR in old animals. The target genes of the circulating miRNAs differentially expressed with age were predicted to regulate pathways relevant to aging, such as cellular metabolism, Wnt signaling, or apoptosis [66]. To Our knowledge, as for DNA methylation, no study has specifically evaluated miRNAs evolution under CR in humans. 

## 4. Histone Modifications

Histones are proteins that form an octamer core particle around which DNA is wrapped, forming the basic structure of the chromatin unit, the nucleosome. Histone tails undergo different types of posttranslational modifications (PTHMs) that can induce changes in chromatin structure. To date, numerous PTHMs have been identified, including methylation, acetylation, ubiquitination, and phosphorylation [67,68], and changes in the abundance and distribution of certain of these specific histone marks have been observed during aging in diverse species [69]. 

Histone modifications are catalyzed by specific enzymes, among which sirtuins, which are nicotinamide adenine dinucleotide (NAD) dependent-HDACs, are known to affect the aging process [70,71]. Sirtuins are present in a variety of organisms, from yeast to mammals, and have been involved in the organization of global chromatin structure (establishment and maintenance of heterochromatin) and in the epigenetic regulation of specific genes. Activation of sirtuins is observed in individuals under CR, in relation with the elevated NAD levels induced by the intervention. Evidence has suggested that the beneficial effects on lifespan extension and healthspan improvement associated with CR are mediated by mechanisms involving sirtuins through (although not exclusively) epigenetic effects [72,73,74]. Data from in vitro models have suggested that sirtuin-dependent, and specially SIRT1-dependent, histone modifications in response to CR could be important mediators of the effects of CR on lifespan, particularly by the regulation of the expression of key genes involved in metabolic pathways, in apoptosis regulation, or in senescence (Table 1). Thus, it was shown by Li et al. that SIRT1 is involved in the regulation of *p16INK4a* gene, a cyclin-dependent kinase inhibitor linked to the regulation of cellular senescence [75]. p16INK4a accumulates during the aging process, characteristic which serves as a robust biomarker of senescence [76]. Using human lung fibroblasts cell lines, the authors demonstrated that glucose restriction resulted in chromatin remodeling of the *p16INK4a* gene promoter. Indeed SIRT1, activated by glucose restriction, could directly bind to the gene promoter and decrease its expression through its deacetylation effects. The decrease in *p16INK4a* expression led ultimately to an inhibition of cellular senescence and to a cellular lifespan extension [75].

Apart from sirtuins, other histone-modifying enzymes are impacted by CR. For example, in the mouse hippocampus, CR prevented the age-related increase in Histone Deacetylase 2 [77], while in yeast (*Saccharomyces cerevisiae*), CR reduced significantly the expression of NAT4, which belongs to the family of acetyltransferases. This decrease of NAT4 induced a loss of histone H4 N-terminal acetylation, subsequently leading to an increase in yeast replicative lifespan [78].

## 5. Impact of Nutritional Interventions on Epigenetic Clocks

During the last six years, there has been tremendous interest in a novel class of biomarkers of aging, called epigenetic clocks. These models, among which Horvath’s [79] and Hannum’s clocks [80] are the most popular ones, combine DNAm values in selected sets of CpG sites in order to predict one individual’s age. Epigenetic clocks are nowadays considered as robust measures of chronological age in humans [81,82,83] but they also appear to be reflective of the biological age of individuals. They can reflect age acceleration associated with the genetic background such as progeroid syndromes [84,85], and deviation of epigenetic age from chronological age (also called epigenetic age acceleration) has been shown to be predictive of important health outcomes such as all-cause mortality or frailty [86,87,88,89,90,91]. Several lines of recent evidence suggest additionally that epigenetic age predictors can be influenced by lifestyle factors and exposures, such as obesity [92,93,94], stress [95,96,97], or pollution [98], but also by nutritional factors. In 2016, Quach et al. were the firsts to investigate the relationships between measures of epigenetic age acceleration and dietary factors [94]. They detected a significant correlation between epigenetic age acceleration measures and fish intake, poultry intake, as well as blood carotenoid levels used as a surrogate of fruit and vegetable consumption [94]. Recently, Levine et al. developed a new powerful biomarker of aging, replacing prediction of chronological age with prediction of a surrogate measure of “phenotypic age” based on several clinical measures [99]. This model, named DNAm PhenoAge, is also related to dietary habits, as indicators of fruits and vegetables consumption are associated with lower values of epigenetic age. Collectively, the results of these two studies suggest that epigenetic clocks are able to grasp the potential health benefits associated with the consumption of higher levels of fruits and vegetables, fish, and poultry [94,99]. 

While studies on the effect of CR on epigenetic clocks in humans are missing, important work has been recently carried out in animal models regarding this question. Indeed, epigenetic clocks have been developed in animal species other than humans, especially in mice and rats [25,46,100,101,102,103,104,105]. These models are of particular interest because rodents are well-known mammalian models of aging and have the potential to provide new insights on the relationships between DNAm biomarkers and interventions (genetic, dietary, or pharmacological) which are intended to increase the lifespan and cannot be easily performed in humans. So far, to our knowledge, six publications have developed epigenetic clock models for rodents: three multi-tissue [100,101,102], two blood-based [25,103], and one liver-based clock [46] (Table 2). With every model, except for the one recently developed by Meer et al. [101], animals treated with CR were significantly younger regarding their epigenetic age compared to their untreated counterparts [25,46,102,103]. For example, CR was able to reduce epigenetic ages of 9.4 months on average in mice livers [46] or of 20% in whole-blood [103]. In Stubbs’ model, the liver predicted epigenetic age was strongly dependent on diet: mice fed with a high-fat diet showed accelerated epigenetic aging, and this phenomenon was even more marked if their mothers were fed with a low-fat diet, underlying the inter-generational effect of DNA methylation [100]. Finally, in the study of Maegawa et al., an epigenetic clock model was also developed for rhesus monkeys: here again, CR was associated with significantly lower methylation age, though the effect was less pronounced than in mice [25].

## 6. Conclusions

With a growing aging population all over the world, healthy aging is an important goal for public health. According to several lines of recent evidence, epigenetic mechanisms occupy a central position in aging and are an attractive area of research as they can be influenced by interventions, such as nutrition. Recent development of validate surrogate biomarkers of aging that rely on DNA methylation data are also a remarkable step forward in aging research. As discussed in this review, nutritional interventions that promote longevity and healthspan could be able to attenuate age-associated epigenetic alterations and could have a protective effect against concomitant cellular alterations. Notably, under CR, the age-related changes in epigenomes are profoundly delayed, as CR promotes maintenance methylation and attenuates the epigenetic drift. As mentioned previously, chronic CR is an intervention difficult to implement in humans. Some authors have proposed intermittent fasting or alternate-day fasting as a surrogate of CR that could extend the lifespan of bacteria, worms, and rodents and could have a positive impact on human healthspan [108,109,110,111]. To our knowledge, no study has evaluated the impact of intermittent fasting on epigenetic signatures of aging. It should be kept in mind that, in humans, nutritional interventions are context-dependent, relying on specific populations, gender, or genetic factors. While some interventions can have beneficial effects in certain individuals, they could, at the same time, be detrimental in other groups. There is a need of gaining insights into the precise molecular mechanisms of action of fasting and, especially, about its impact on epigenetic reprogramming. The development and testing of interventions that aim to achieve more successful aging are current exciting goals in research that should be pursued in humans without being harmful [109]. 

## Figures and Tables

**Table 1 ijms-20-02022-t001:** Impact of caloric restriction on age-associated microRNAs (miRNAs) changes and histones modifications.

Study	Model	Intervention	Biological Matrix/Tissue	Impact of Intervention
**miRNAs**				
Khanna et al. 2011 [63]	Mice	CR mice in three age groups: 12, 24, and 28 months.	Brain	Under CR, no age-dependent up-regulation of miR-181a-1*, miR-30e, and miR-34a, as observed in AL fed animals, associated with a gain in the expression of their target gene *Bcl-2*.
Mercken et al. 2013 [65]	Rhesus monkeys (all males)	Animals maintained on CR diet (TestDiet® #5L1F, Purina Mills) most of their lives (20.8–22.6 years).	Skeletal muscle	CR was able to reverse the age-related alterations in miRNA expression.
Dhahbi et al. 2013 [66]	Mice (B6C3F1 strain)	CR (<40% fewer calories than the control group) from 1 month until 27 months of age.	Serum	CR antagonized the increase in serum levels of a large set of miRNAs.
Wood et al. 2015 [35]	Rats	55% CR until sacrifice (6, 12, or 28 months).	Cerebral cortex	Significant overexpression of miR-98-3p in all groups of CR rats.
**Histones modifications**				
Li et al. 2011 [75]	Normal diploid WI-38, MRC-5 and IMR-90 human fetal lung fibroblasts	Glucose restriction (glucose- and pyruvate-free DMEM medium)	NA	Activation of SIRT1 by glucose restriction led to chromatin remodeling of the *p16INK4a* gene promoter and a decreased expression of this gene, ultimately associated with the inhibition of cellular senescence.
Chouliaras et al. 2013 [77]	Mice (males) (C57Bl6J wild-type strain and transgenic animals overexpressing SOD1)	50% CR until sacrifice (12 or 24 months).	Hippocampus	CR prevented the age-related increase in histone deacetylase 2 (HDAC2) levels.
Molina-Seranno et al. 2016 [78]	Yeast (*Saccharomyces cerevisiae*)	Reduction of glucose concentration in growth media from 2 to 0.1%.	NA	CR was associated with a reduction of histone H4 N-terminal acetylation.

CR: caloric restriction; AL: ad-libidum; NA: not applicable.

**Table 2 ijms-20-02022-t002:** Epigenetic clocks in animal models.

Study	Animal Model	DNA Methylation Analysis	Intervention	Biological Matrix/Tissue	Number of CpG Sites	Impact of Diet or CR
Maegawa et al. 2017 [25]	Mice and Rhesus macaques	DREAM Methylation analysis [106]	*Mice*: 40% CR starting at 0.3 years of age until 2.7–3.2-year-old. *Rhesus macaques*: 30% CR starting in middle age (age: 7–14 y) and analyzed at 22–30 years of age (CR treatment period: 15–21 years).	Blood	24	Animals under CR had significantly lower epigenetic predicted ages compared to their chronological age.
Wang et al. 2017 [46]	Female mice (UM-HET3 mice)	RRBS	60% CR until 22-months-old	Liver	148	Reduction of epigenetic age of 9.4 months on average in CR mice versus their age-matched controls.
Stubbs et al. 2017 [100]	Male mice (C57BL/6J)	RRBS	Low-fat–high-carbohydrate diet or high-fat–low carbohydrate diet	Multi-tissue predictor applied to liver samples [107]	329	Animals on high-fat diet showed accelerated epigenetic aging.
Petkovich et al. 2017 [103]	Male mice (C57BL/6J and B6D2F1)	RRBS	Dietary intervention started at 14 weeks for all mice, until 10, 18, 23, or 27 months (C57BL/6J mice) or until 21 or 27 months (B6D2F1 mice).	Multi-tissue	90	On average, mice under CR had an epigenetic age 20% lower than their chronological age. The effect was less pronounced in younger animals than in older ones.
Thompson et al. 2018 [102]	Mice (use of publicly available datasets)	RRBS [103] and WGBS [42]	*WT UM-HET3 mice*: CR diet initiated at 4 months of age until 22 months of age [42]. *C57BL/6* and *B6D2F1* mice: see above Petkovich et al. [103]	Multi-tissue	Development of four new models of epigenetic clock (elastic net clock, ridge regression clock, two conserved clocks)	Delayed epigenetic aging effects observed in C57BL/6, B6D2F1, and HET3 mice under CR.
Meer et al. 2018 [101]	Male mice (C57BL/6J and B6D2F1)	RRBS	*C57BL/6* and *B6D2F1* mice: see above Petkovich et al. [103]	Multi-tissue	435	Shift towards a younger age in animals under CR, but difference not statistically significant.

RRBS: reduced-representation bisulfite sequencing; WGBS: whole-genome bisulfite sequencing.

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
