# Peer review of "The Impact of Caloric Restriction on the Epigenetic Signatures of Aging"

_ijms, 2019, doi:10.3390/ijms20082022_

Round 1

Reviewer 1 Report

In this manuscript, the authors reviewed how CR influences epigenetic signatures of aging in terms of DNA methylation, miRNA expression, histone modifications, and alterations of epigenetic clocks. It is a thoroughly written review on this topic.  I recommend it to be accepted with minor language editing.

Author Response

We thank the reviewer for the comments.

A language editing has been performed.

Reviewer 2 Report

The review "The impact of Caloric Restriction on Epigenetic Signature of Aging" by Gensous N. et al is a comprehensive review of the current knowledge regarding the effect of caloric restriction on the epigenetic modifications in pathways related to aging.

Comments:

Table 1. summarizes the effects of DNA methylation studies and it would be helpful for the readers if the authors would include a summary of the miRNAs and/or Histone modifications, and/or nutritional interventions.

Also, it would be advisable if the authors would consider for Table 1. an alternative word to Tissue in the 5th column and include biological matrix/ tissue given that blood was used in one of the studies discussed.

Author Response

We thank the reviewer for the useful comments.

Point 1 : Table 1. summarizes the effects of DNA methylation studies and it would be helpful for the readers if the authors would include a summary of the miRNAs and/or Histone modifications, and/or nutritional interventions.

A new table entirely devoted to miRNAs and histone modifications has been included in the text (new Table 1).

Point 2 : Also, it would be advisable if the authors would consider for Table 1. an alternative word to Tissue in the 5th column and include biological matrix/ tissue given that blood was used in one of the studies discussed.

The header of the table (now Table 2) has been updated as suggested.